# Age-Related Variation in the Provision of Primary Care Services and Medication Prescriptions for Patients with Cardiovascular Disease

**DOI:** 10.3390/ijerph191710761

**Published:** 2022-08-29

**Authors:** Qiang Tu, Karice Hyun, Nashid Hafiz, Andrew Knight, Charlotte Hespe, Clara K. Chow, Tom Briffa, Robyn Gallagher, Christopher M. Reid, David L. Hare, Nicholas Zwar, Mark Woodward, Stephen Jan, Emily R. Atkins, Tracey-Lea Laba, Elizabeth Halcomb, Tim Usherwood, Laurent Billot, Julie Redfern

**Affiliations:** 1Faculty of Medicine and Health, School of Health Sciences, The University of Sydney, Sydney 2050, Australia; 2Department of Cardiology, Concord Hospital, Sydney 2139, Australia; 3The Primary and Integrated Care Unit, South Western Sydney Local Health District, Sydney 2170, Australia; 4School of Population Health, University of New South Wales, Sydney 2052, Australia; 5School of Medicine, The University of Notre Dame, Sydney 2010, Australia; 6Research Education Network, Western Sydney Local Health District, Sydney 2151, Australia; 7Westmead Applied Research Centre, Faculty of Medicine and Health, The University of Sydney, Westmead 2154, Australia; 8School of Population and Global Health, The University of Western Australia, Perth 6009, Australia; 9Faculty of Medicine and Health, Sydney Nursing School, University of Sydney, Sydney 2006, Australia; 10School of Population Health, Curtin University, Perth 6102, Australia; 11School of Public Health and Preventive Medicine, Monash University, Melbourne 3004, Australia; 12Faculty of Medicine, Dentistry and Health Sciences, University of Melbourne, Melbourne 3010, Australia; 13Department of Cardiology, Austin Health, Heidelberg 3084, Australia; 14Faculty of Health Sciences & Medicine, Bond University, Gold Coast 4226, Australia; 15The George Institute for Global Health, University of New South Wales, Sydney 2046, Australia; 16The George Institute for Global Health, School of Public Health, Imperial College London, London NW9 7PA, UK; 17Pharmacy Program, Clinical and Health Sciences Unit, University of South Australia, Adelaide 5001, Australia; 18School of Nursing, University of Wollongong, Wollongong 2522, Australia

**Keywords:** cardiovascular disease, age, secondary prevention, primary care, risk factors

## Abstract

As population aging progresses, demands of patients with cardiovascular diseases (CVD) on the primary care services is inevitably increased. However, the utilisation of primary care services across varying age groups is unknown. The study aims to explore age-related variations in provision of chronic disease management plans, mental health care, guideline-indicated cardiovascular medications and influenza vaccination among patients with CVD over differing ages presenting to primary care. Data for patients with CVD were extracted from 50 Australian general practices. Logistic regression, accounting for covariates and clustering effects by practices, was used for statistical analysis. Of the 14,602 patients with CVD (mean age, 72.5 years), patients aged 65–74, 75–84 and ≥85 years were significantly more likely to have a GP management plan prepared (adjusted odds ratio (aOR): 1.6, 1.88 and 1.55, respectively, *p* < 0.05), have a formal team care arrangement (aOR: 1.49, 1.8, 1.65, respectively, *p* < 0.05) and have a review of either (aOR: 1.63, 2.09, 1.93, respectively, *p* < 0.05) than those < 65 years. Patients aged ≥ 65 years were more likely to be prescribed blood-pressure-lowering medications and to be vaccinated for influenza. However, the adjusted odds of being prescribed lipid-lowering and antiplatelet medications and receiving mental health care were significantly lowest among patients ≥ 85 years. There are age-related variations in provision of primary care services and pharmacological therapy. GPs are targeting care plans to older people who are more likely to have long-term conditions and complex needs.

## 1. Introduction

Cardiovascular disease (CVD), including coronary heart disease, myocardial infarction, hypertension, heart failure, stroke, peripheral vascular disease, carotid stenosis and renal artery stenosis, is the leading cause of morbidity and mortality globally, which dominates healthcare and challenges health systems [1]. The world’s population aged 60 years and over is projected to increase from 1 billion in 2020 to 2.1 billion by 2050 [2]. There are a number of underlying risk factors for developing CVD, which include non-modifiable (age, ethnicity, gender, genetics) and modifiable risk factors (elevated blood pressure, diabetes, smoking, overweight, physical inactivity, excessive alcohol consumption and underuse of prescription medications) [3]. To achieve the optimal management of CVD risk and reduce the burden of established CVD, the European Society of Cardiology guideline advocates implementing secondary prevention of CVD in primary care [4]. It is widely recognised by the World Health Organisation and reflected in international guidelines that effective secondary prevention strategies for CVD include vigorous management, monitoring and treatment to target of risk factors, appropriate treatments with multiple evidence-based cardiovascular medications and timely receipt of chronic care services in primary care [5,6].

Australia has a universal health care insurance system (Medicare), which is the primary funding mechanism for subsidising primary care services and pharmaceuticals for Australian citizens and residents [7]. Under Medicare, the Australian health system aims to achieve its core objectives enabling patients to have access to appropriate primary care services and medications [8,9]. Medicare comprises two elements: the Medical Benefits Schedule (MBS) and the Pharmaceutical Benefits Scheme (PBS). The MBS includes Chronic Disease Management Plan (CDMP) items and GP Mental Health Care items. CDMP items provide rebates for (i) general practice management plans (GPMPs) to foster systematic care planning, (ii) team care arrangements (TCAs) to improve multidisciplinary-based integrated care, and (iii) reviews of both GPMP and TCA to support continuity of care [10]. As part of MBS-subsidised items, GP mental health care items provide rebates for (i) assessment and preparation of a mental health management plan, (ii) review of the management plan, and (iii) mental health treatment consultations [11]. Under the PBS, the government subsidises the cost of evidence-based cardiovascular medications including blood pressure-lowering, lipid-lowering, and antiplatelet medications [12,13]. Evidence has demonstrated that the utilisation of MBS- and PBS-subsidised health services and medications are associated with a substantial reduction in cardiovascular hospitalisation and premature mortality [14]. 

Despite widespread efforts made in the secondary prevention of CVD in primary care through universal health coverage, gender and ethnicity-related variations persist. Areas of variability include utilisation of primary care services such as chronic disease management, mental health care, cardiovascular medications, and influenza vaccination [15,16,17]. As population aging progresses, demands on the existing primary care services will be inevitably increased and require more knowledge about the utilisation of primary care services across varying age groups [18]. Studies have suggested that age is one of the key determinants that influence health-seeking behaviour in primary care [19,20]. However, there have been limited studies that have examined whether usage of primary care services and prescribed medications vary by age [21,22,23,24,25]. Most previous studies in assessing age–related differences in health services use treated patients aged 65 years and above as a single homogeneous group [21], lack focus on a specific disease condition (e.g., CVD) [23,24], or do not consider secondary prevention at the primary care level [22,25]. Therefore, the aim of this study was to evaluate age-related variation in the utilisation of primary care services including CDMPs, mental health care, prescription of guideline-indicated medications and influenza vaccination among patients with CVD presenting to primary care.

## 2. Materials and Methods

A retrospective cross-sectional study was conducted using data from the QUEL study an Australia–based cluster randomised controlled trial that recruited 50 primary care practices across four states, including New South Wales, Victoria, South Australia and Queensland [26]. Ethics approval has been obtained from the New South Wales Cancer Institute Population and Health Services Research Ethics Committee (no. HREC/18/CIPHS/44). Waiver of participant consent has been granted for this baseline data analysis due to the characteristics of extraction of de-identified data from electronic health record system. Given it is a practice-level study, participating general practices have consented to provide de-identified data for analysis.

### 2.1. Practices and Participants

Primary care practices were eligible if they (i) were located in Australia, (ii) managed ≥ 100 patients with CVD annually, and (iii) installed compatible data extraction software. Inclusion criteria for patients (i) were aged ≥ 18 years, (ii) a documented diagnosis of CVD as recorded by the general practitioner including coronary heart disease, acute coronary syndrome, myocardial infarction, heart failure, stroke, and peripheral vascular disease, and (iii) had visited the participating practice ≥ 3 times in the previous 24 months. Inclusion of patients was based on data entered into electronic medical records by treating primary care providers at each practice and hence diagnosis was a clinical record and not verified by source documentation.

### 2.2. Data Collection

Data extraction was conducted by an external population health analytics and reporting platform named Pen Computer Systems (Pen CS) [27]. De-identified data between 2018 and 2019 were automatically extracted from 50 participating primary care practices. Extracted data were stored on the university computer server under password protection. To ensure privacy, the practice name was replaced by a specific practice number during data extraction and a statistician who was blinded to the practice number performed data pooling and analysis. A range of demographic and clinical data were extracted including age, gender, indigenous status, CVD risk factors (blood pressure, total cholesterol, high-density lipoproteins, low-density lipoproteins, glycated haemoglobin (HbA1c), smoking status, alcohol consumption, body mass index (BMI)), mental health diagnoses, prescribed medications, MBS items and influenza vaccination. Three types of guideline-indicated cardiovascular medications prescribed between 2018 and 2019 were also extracted. This included blood pressure-lowering (beta-blockers, angiotensin-converting enzyme inhibitors, angiotensin receptor blockers, calcium channel blockers and diuretics), lipid-lowering, and antiplatelet medications. In addition, claimed MBS items were extracted to enable age-related comparison in the receipt of both CDMPs and mental health care. Specific descriptions for MBS items are presented in Table 1. The extracted data were securely uploaded to the university’s research data storage platform for analysis.

### 2.3. Outcomes

The primary outcome of interest was the uptake of CDMP by patients with CVD (Table 1). Secondary outcomes included (1) mental health claims among patients with CVD who were documented as having a mental health disorder, (2) medication prescription (blood pressure-lowering, lipid-lowering, and antiplatelet medications) written by GPs among patients with CVD, and (3) influenza vaccination. Specific descriptions for each MBS item are detailed in Table 1 [28].

### 2.4. Statistical Analysis

Patients were stratified into four age groups: adults < 65 years, youngest-old (65–74 years), middle-old (75–84 years), and oldest-old (≥85 years). Categorical variables were summarised as number and percentage and continuous variables were summarised as mean and standard deviation (SD) after visually checking that the distribution is normal. Cochran-Armitage trend test and univariable linear regression were used to test for the univariable trend of binary and continuous patient characteristics and outcomes, respectively, across the increasing age groups. For adjusted analyses, multivariable logistic regression models were performed using generalised estimating equations to account for the clustering effect of practices. We included the following independent variables in the models: age (65–74, 75–84, ≥85 vs. <65 years), gender (male vs. female), Indigenous status (yes vs. no), current smoker (yes vs. no), alcohol drinker (yes vs. no), type 2 diabetes (yes vs. no), BMI, systolic blood pressure, and total cholesterol. Based on the clinical significance, risk factors for CVD were included in the models as covariates to adjust for the severity of cardiovascular risk in patients. The adjusted odds ratio (aOR) and 95% confidence intervals (95% CIs) were estimated to present the aged-related variations. When analysing the receipt of mental health items, only patients with CVD who were documented as having a mental health disorder were included. A two-sided *p*-value of < 0.05 was considered statistically significant. All statistical analyses were undertaken using SAS version 9.4.

## 3. Results

### 3.1. Patient Characteristics

A total of 14,602 patients with CVD were included in the study. Demographic and clinical characteristics of patients, stratified by age, are presented in Table 2. There were 8842 (61%) male patients. The mean age was 72.5 (SD 12.7) years: 3666 (25%) patients were aged < 65 years, 4229 (29%) patients were aged 65–74 years, 4108 (28%) patients were aged 75–84 years and 2599 (18%) patients were aged ≥ 85 years. In total, 3078 (21%) patients with CVD had coexisting type 2 diabetes. Compared with those < 65 years, patients with CVD in all age categories ≥ 65 years (65–74, 75–84 and ≥ 85 years) were significantly less likely to be a current smoker (13%, 6%, 2%, respectively, vs. 22%, *p* < 0.0001), alcohol drinker (60%, 55%, 48%, respectively, vs. 63%, *p* < 0.0001), or have type 2 diabetes (22%, 24%, 19%, respectively, vs. 17%, *p* < 0.0001).

Mean systolic and diastolic blood pressure of the total cohort were 132.1 mmHg and 74.9 mmHg, respectively. Statistically significant differences in the systolic and diastolic blood pressure were found among the four age groups where age–related increase in systolic blood pressure and decrease in diastolic blood pressure were noted. Patients with CVD aged < 65 years had the highest prevalence of mental health disorder compared with other three age groups.

### 3.2. Provision of Chronic Disease Management Plans

Of the total cohort, the proportion of patients with CVD who received ‘preparation of GPMP’, ‘coordination of TCA’ and ‘review of GPMP or TCA’ were 44%, 38% and 35%, respectively (Table 3). In particular, for the patient group < 65 years, the proportion of receiving GPMP, TCA and review was only 35%, 30% and 26%, respectively. The proportion of patients < 65 years who had both GPMP and review was lower, being only at 22%.

Significant age-related differences in the receipt of all three sub–types of CDMPs were found such that an inverted U-shaped relationship with age in utilisation of GPMP, TCA and review occurred. The utilisation rate of GPMP, TCA and review was lower in the extreme age groups (<65 and ≥85 years) than in the middle two age groups (65–84 years). After adjusting for demographic and clinical factors, the difference in the proportion among the age groups receiving these three sub-types of CDMPs remained significant (Table 4). Patients with CVD aged 65–74, 75–84 and ≥85 years were significantly more likely to have a GPMP prepared (aOR: 1.6, 1.88 and 1.55, respectively, *p* < 0.05), have a TCA (aOR: 1.49, 1.8, 1.65, respectively, *p* < 0.05) and have a review of GPMP or TCA (aOR: 1.63, 2.09, 1.93, respectively, *p* < 0.05) than those < 65 years.

### 3.3. Provision of Mental Health Care

In total, 25% of patients with CVD were documented as having a mental health disorder. Patients aged < 65 years (28%) were significantly more likely to be documented as having a mental health disorder compared with those aged 65–74 (25%), 75–84 (22%), and ≥ 85 years (24%) (*p* < 0.0001). Increasing age was associated with a lower proportion receiving a GP mental health treatment plan, review of said plan, and documentation regarding mental health treatment consultation. The adjusted odds of receiving mental health treatment consultation (0.69, 0.69 and 0.49, respectively, *p* < 0.05), preparation of GP mental health care plan (0.55, 0.3 and 0.17, respectively, *p* < 0.05), and review of a GP mental health treatment plan (0.51, 0.34 and 0.11, respectively, *p* < 0.05) was significantly lower amongst patients aged 65–74, 75–84 and ≥85 years in comparison with those aged < 65 years (Table 4). Patients ≥ 85 years who were provided ‘preparation of a GP Mental Health Treatment Plan’, ‘Review of a GP Mental Treatment Plan’, and ‘Mental health Treatment Consultation’ were only 3%, 1% and 5%, respectively.

### 3.4. Prescription of the Guideline-Recommended Cardiovascular Medications

The proportion of patients prescribed evidence–based medications was lower in the oldest and youngest age groups (aged < 65 and aged ≥ 85 years) than in the middle age groups (65–84 years old) (Table 3). Patients aged ≥ 85 years had fewer prescriptions of combination cardiovascular medication than the younger age groups. Patients aged 75–84 years had the highest proportion (74%) of guideline-indicated blood pressure–lowering medication prescriptions while patients aged < 65 years had the lowest (59%). Patients aged 65–74 years were prescribed more (63% and 52%, respectively) lipid–lowering and antiplatelet medications than the other three groups, while patients aged ≥ 85 years had the lowest prescription of lipid-lowering and antiplatelet medications (44% and 41%). After adjustment for clinical and demographic factors, the age differences in the prescription of these three types of medications remained significantly different. Patients aged 75–84 years had 1.9 times the odds of receiving blood pressure-lowering medication than those aged < 65 years. Patients aged 65–74 had a more than 96% and 38% higher likelihood of being prescribed lipid-lowering medication and antiplatelet than those aged ≥ 85 years, respectively. Further, prescription of lipid–lowering medications and antiplatelet medications across all four age groups were consistently less frequent than the prescription of blood pressure–lowering medications.

### 3.5. Influenza Vaccination

Seventy-five percent of the cohort were vaccinated against influenza. Patients aged 75–84 years had the highest proportion of receiving influenza vaccination while those aged < 65 years had the least. The adjusted odds of receiving influenza vaccination were significantly higher amongst patients aged 65–74, 75–84 and ≥85 years (aOR; 3.13, 6.19 and 8.23, respectively, *p* < 0.05) compared with those aged < 65 years.

## 4. Discussion

To the best of our knowledge, this is the first study to comprehensively evaluate age-related variations in the provision of primary care services and guideline-indicated prescribed medications among patients with CVD. Overall, we found substantial age-related variation in the provision of CDMPs, mental health care, guideline-indicated prescribed medications, and influenza vaccination, even after adjusting for social-demographic and clinical factors. 

Age-related variation in provision of Medicare-funded CDMP items was found in patients with CVD who attended primary care practices. Our study found that CDMP items were not commonly claimed by GPs treating patients with CVD, although absence of claiming does not necessarily reflect that chronic disease support is not occurring. It revealed an overall underutilisation of CDMP for patients with CVD, particularly for the younger cohort who might benefit the most from these item numbers due to the associated subsidised access to allied health services. Previous studies have shown that increased use of GPMP was associated with reduced hospitalisation in older patients with chronic conditions [14,29]. Therefore, our results are noteworthy and deserve further investigation as to why these items were not commonly used or claimed. Understanding the causes of variation is important to develop targeted quality improvement programs. Our findings in CDMP items generally support an increasing utilisation of GPMP, TCA and a review with advancing age before 85 years. Given that older people were more prone to more chronic conditions and multimorbidity than their younger counterparts, it is difficult to infer whether increased use of CDMPs equate to better care for CVD. However, increased CDMP use may imply a positive finding suggesting that the GPs are appropriately targeting care plans and providing chronic disease management where they think it is likely to have most benefit for patients. The increased use of CDMP items observed may also reflect the patients’ desire to access allied health funding for competing chronic conditions and thus represents an appropriate response to greater needs for chronic care along with increasing age. However, the utilisation rate of CDMP items declined moderately in the age group of ≥ 85 years, generating an overall inverted U–shaped relationship with age. This trend may be explained by the life expectancy theory. In Australia, the average life expectancy was 82.8 years [30]. As health care utilisation often peaks at the end of life, it partly explained our finding that the highest proportion of using chronic disease management plans occurred in the 75–84 age group and subsequently a moderate decrease in the ≥85 age group [31]. The relatively lower percentage of generating CDMPs in the ≥85 age group may be associated with time and resource constraints, substitution of institutional care, functional decline or frailty of older people impeding their ability to seek healthcare, patient preference near the end of life [24]. Overall, there is a potential opportunity to increase the quality of healthcare in patients with CVD by better utilisation of proactive healthcare planning and reviews.

With respect to provision of mental health care, our results showed an age-related variation. The present study found that, among patients with comorbid CVD and mental health disorders, preparation of GP mental health treatment plans, review of a GP mental treatment plan and mental health treatment consultation steadily decreased with increasing age, from 22%, 9%, 15% in age group of <65 years to 13%, 5%, 1% in age group of ≥85 years. Our results differ from previous research that found the elderly had more positive healthcare-seeking attitudes regarding mental health concerns than younger patients [20]. Potential reasons explaining the age-related shift in mental health service use in our study may include prioritisation of physical health care above mental health care for older people, the acknowledgment of mental health disorders as a normal or inevitable part of aging process, perceived stigma, or discrimination for seeking mental health treatment, or poor mental health literacy of older patients [32,33]. Qualitative review of these findings would assist in addressing the gap in care and therefore improving both health outcomes and quality of life. Mental health disorders, such as depression, can lead to poor health behaviours including unhealthy diet, smoking, physical inactivity, and medication nonadherence [34]. Poor health behaviours act as barriers against secondary prevention of CVD and contribute to adverse CVD outcomes [21]. Therefore, our results indicate it is necessary to continue to optimise mental health care among patients with CVD in primary care settings through identification, assessment and treatment of mental health issues and integration of mental health into chronic disease treatment.

Our results also revealed the existence of significant variation across age-groups in the prescription of guideline-directed CVD medications in Australia. This finding is consistent with a previous Scottish population-based cross-sectional study, which found that people aged 65–84 years were prescribed more evidence-based CVD medications than those ≥ 85 years [35]. The finding that relatively less prescription of lipid-lowering medication and antiplatelet medication among patients aged ≥ 85 years than those aged 65–84 years observed in our study is also consistent with previous studies. An Australian study showed that only 3% of centenarians received lipid-lowering medications compared to 49% of those aged 65 to 74 [23]. Another study from the Netherlands found that a continuous increase in the use of lipid-lowering medications in patients with CVD up to the age of 85 years after which its use markedly declined [12]. In addition, a retrospective cohort study in the USA has found that patients older than 75 years with myocardial infarction were less likely to be prescribed aspirin than those aged 65 to 74 years [36]. The current study also showed that among older patients aged > 65 years, combination of blood pressure–lowering medication, lipid-lowering medication and antiplatelet medication gradually declined with increasing age. It is difficult to interpret whether the decreased prescription at older ages reflects clinical judgement, limitations of PBS, patient-centred clinical decision making, suitability of CDMP or patient preference. It may be partly attributed to a treatment-risk paradox, whereby GP’s concerns on comorbidities, age-related side effects and intolerance of medications, contraindications for pharmacological therapy of oldest-old population due to their changing physiological characteristics [12,21,37,38]. Therefore, based on safety considerations, GPs and patients may tend toward conservative treatment aligned with a philosophy of “de-prescribing”. Further, our study found that prescription of lipid-lowering (44%) and antiplatelet medications (41%) were much lower than blood pressure-lowering medications (69%) among patients aged ≥ 85 years. However, the gap was smaller among the other three age groups. Similar results were shown in a previous study that the use of blood pressure-lowering medications among patients ≥ 75 with coronary heart disease was more than three times higher than lipid-lowering medications [12]. These findings further highlight the GP’s demand of stronger worded CVD guidelines for prescribing preventive care in the older years and importance of age-specific consideration in prescribing CVD medications by GPs, particularly for older people who have competing medical needs and are the users of polypharmacy.

It was not surprising to see the proportion receiving influenza vaccination increased with age, given people aged ≥ 65 years are eligible for free influenza vaccination in Australia, whereas those aged < 65 years need a specific indication. This positive finding was aligned with the recommendation of national immunisation program. A systematic review and meta-analysis has found that influenza vaccination was associated with a lower risk of cardiovascular mortality and adverse cardiovascular events [39]. Therefore, secondary CVD care needs to be funded for influenza vaccination under the age of 65 and policy makers should consider adding them to the priority group. Consideration should also be given to including secondary prevention of CVD as an indication in the subsidised influenza vaccination scheme.

The reasons behind the observed variations may be complex. It is important to raise awareness among health providers that optimal management of patients with CVD with careful age consideration is crucial to improve outcomes. Understanding these variations more thoroughly may help improve the appropriate utilisation of MBS–subsidised health services. Further research into the reasons for the variations is needed, for example, an in-depth exploration of health providers’ attitudes about the care of older patients and patients’ experiences in receiving polypill treatments and primary care services and reimbursement of care plans. This may provide important insights on how to inform current policy development and facilitate the improvement of implementation of treatment options in primary care. Further, to maximise benefits of primary care, we suggest health services customised to the age-specific care needs of patients are provided and better guidelines around appropriate preventive care for older people are developed.

## 5. Strengths and Limitations

The major strength of our study includes the large sample size and the automatic extraction of individual patient data which ensures reliability and integrity of data and minimises the potential sampling bias and inaccuracies of self-reported data. The large cohort of patients from 50 practices across four states increased the generalisability of the findings. However, there are also limitations. First, not all sociodemographic data (e.g., education, and the severity and duration of diseases) were available in the database. Despite controlling for several confounders, residual confounding likely persists. Second, we had no information on the indication for prescription, or reasons for not prescribing such as contraindication or medication intolerance. Therefore, age-related variation in medication treatment might be a consequence of patients’ clinical symptoms or differences in risk estimates. Third, the overall use of medications is unclear. The extracted data were written prescriptions and did not include non-prescription or over-the-counter medications. Written prescriptions do not guarantee patients took the medications. Fourth, each patient’s clinical diagnosis was extracted from electronic health records entered by primary care providers. Hence, diagnosis was not verified by source documentation. Further, diagnoses may not have been recorded correctly and so not extracted. Patients with a mental health diagnosis may receive influenza vaccination at the pharmacy and not have it recorded in their GP’s electronic health record. Fifth, health service use may be underestimated as the absence of claiming items does not reflect care support is not occurring. Sixth, this was a secondary analysis and results are based on an Australian cohort presenting to primary care, which may limit generalizability. Further, the data analysed were the baseline data of a cluster randomised controlled trial and were not collected specifically to answer the research questions. Seventh, considering the small proportion of young patients with CVD included, a comparison between young age and old age was not conducted. Future study with a large number of young people needs to be conducted to compare the item uses between young age and old age. Further, gender sub-analysis was not performed. Age variations between male and female could be the subject of future work. The limitations of this study must be considered in interpreting the results. Despite the limitations, the analysis provides a useful picture of real-world patterns of health services and medication uses in primary care practices among CVD patients. The results are not anticipated to reflect the level of quality care.

## 6. Conclusions

Age-related variations emerged in terms of utilisation of CDMPs, mental health care, cardiovascular medications, and influenza vaccination in participating primary care practices. GPs are appropriately targeting care plans for patients who may gain most benefit and are in most healthcare need. CVD patients < 65 years were less likely to receive CDMPs and influenza vaccination but more likely to receive mental health care MBS items than older patients. Patients < 65 years had the lowest odds of being prescribed blood-pressure-lowering medications while patients aged ≥ 85 had the lowest odds of being prescribed lipid-lowering medications and antiplatelet medications. Age-related variations found in our study may have implications for future secondary prevention strategies in primary care targeting age-specific care needs and efficient allocation of health resources. The observed differences by age offer an opportunity to optimise CVD in primary care in all age groups to address the escalating CVD burden. Ongoing efforts are warranted to ensure improved uptake and delivery of primary care services that are appropriately targeting those in most need. Further research will help deepen understandings of these variations in care.

## Figures and Tables

**Table 1 ijerph-19-10761-t001:** Medicare Benefits Schedule for chronic disease management and mental health care in general practice and Pharmaceutica Benefits Scheme.

Australia Universal Insurance System	Elements	Item Name and Number		Rebate Amount (AUD)	Description
Medicare	Medical Benefits Schedule (MBS)	Chronic Disease Management Plan (CDMP) Items	721 Preparation of a General Practice Management Plan (GPMP)	$148.75	Rebate for GP to prepare a management plan for patients with a chronic or terminal medical condition
			723 Coordination of Team Care Arrangements (TCAs)	$117.90	Rebate for GP to develop management plan for patients with a chronic or terminal medical condition and complex needs requiring ongoing care from a multidisciplinary team (at least three care providers)
			732 Review of a GPMP and/or TCA	$74.30	Rebate for GP to review a GPMP and/or TCA
		Mental Health Care Items	2700/2701/2715/2717 Preparation of a GP Mental Health Treatment Plan -	$73.95/$108.85/$93.90/$138.30	Rebate for a GP to assess patients and prepare a care management plan for patients who have mental illness. Different item numbers indicate the varying duration in service provision and whether the GP has undertaken mental health skills training.
			2712 Review of a GP Mental Health Treatment Plan	$73.95	Rebate for a GP to review patients with a GP Mental Health Treatment Plan.
			2713 GP Mental Health Treatment Consultation	$73.95	Rebate for a GP for the ongoing management of patients with mental health disorder including patients being managed with or without a GP Mental Health Treatment Plan.
	Pharmaceutica Benefits Scheme (PBS)	With the use of PBS, the government subsidises the cost of evidence-based cardiovascular medications including blood pressure-lowering, lipid-lowering, and antiplatelet medications			

**Table 2 ijerph-19-10761-t002:** Demographic and clinical characteristics of the cohort with established CVD by age.

Variable	<65 (*n* = 3666)	65–74 (*n* = 4229)	75–84 (*n* = 4108)	≥85 (*n* = 2599)	Total (*n* = 14602)	Data Available *n* (%)	*p*-Value
Male, *n* (%)	2453 (67)	2766 (65)	2477(60)	1146 (44)	8842 (61)	14,601 (99)	<0.0001
Age, mean (SD) (years)	55.6 (7.61)	69.9 (2.83)	79.3(2.88)	89.8(4.04)	72.5 (12.72)	14,602 (100)	<0.0001
Indigenous, N (%)	169 (5)	101 (3)	55 (2)	15 (1)	340 (3)	12,418 (85)	<0.0001
Diagnosis of diabetes, *n* (%)							
Type 1	65 (2)	63 (1)	22 (1)	13 (1)	163 (1)	14,602 (100)	<0.0001
Type 2	640 (17)	940 (22)	998 (24)	500 (19)	3078 (21)	14,602 (100)	0.001
Cardiovascular risk factors							
Systolic blood pressure (mm Hg), mean (SD)	129.5 (16.82)	132.4 (17.13)	133.0 (18.25)	133.6 (20.02)	132.1 (17.97)	13,968 (96)	<0.0001
Diastolic blood pressure (mm Hg), mean (SD)	79.7 (29.02)	75.1 (23.50)	72.3 (30.20)	71.6 (42.11)	74.9 (30.78)	13,976 (96)	<0.0001
Total cholesterol (mmol), mean (SD)	4.5 (1.22)	4.2 (1.09)	4.1 (1.07)	4.3 (1.15)	4.2 (1.14)	13,309 (91)	<0.0001
High-density lipoprotein (mmol/L), mean (SD)	1.3 (0.37)	1.3 (0.39)	1.3 (0.41)	1.4 (0.40)	1.3 (0.39)	12,761 (87)	<0.0001
Low-density lipoprotein (mmol/L), mean (SD)	2.4 (1.03)	2.2 (0.93)	2.1 (0.88)	2.2 (0.95)	2.2 (0.95)	12,634 (87)	<0.0001
Body mass index, mean (SD)	31.1 (11.63)	30.3 (11.36)	28.8 (7.93)	26.8 (10.04)	29.5 (10.41)	11,633 (80)	
Body mass index > 24.9 kg/m^2^, N (%)	2088 (75)	2616 (77)	2471 (73)	1179 (61)	8354 (73)	11,522 (79)	<0.0001
Current smoker, N (%)	753 (22)	494 (13)	220 (6)	49 (2)	1516 (11)	13,250 (91)	<0.0001
Alcohol drinker, N (%)	1562 (63)	1787 (60)	1593 (55)	784 (48)	5726 (57)	10,013 (69)	<0.0001
HbA1c for those with diabetes, mean (SD)	7.0 (5.24)	7.8 (7.66)	7.6 (7.10)	8.1 (8.84)	7.6 (7.23)	8337 (57)	<0.0001
Achieved cardiovascular risk factor targets							
Systolic blood pressure < 130 mm Hg	1763 (50)	1708 (42)	1651 (42)	992 (41)	6114 (44)	13,968 (96)	<0.0001
Diastolic blood pressure < 80 mm Hg	1775 (51)	2673 (65)	2919 (74)	1840 (76)	9207 (66)	13,976 (96)	<0.0001
High-density lipoprotein > 1.0 mmol/L	2163 (70)	2831 (74)	2805 (75)	1698 (81)	9497 (74)	12,761 (87)	<0.0001
Low-density lipoprotein < 1.8 mmol/L	892 (29)	1452 (38)	1524 (41)	747 (36)	4615 (37)	12,634 (87)	<0.0001
Total cholesterol < 4.0 mmol/L	1267 (39)	1920 (49)	1964 (51)	1067 (46)	6218 (47)	13,309 (91)	<0.0001
HbA1c ≤ 7%	1440 (75)	1913 (76)	1980 (79)	1106 (81)	6439 (77)	8337 (57)	<0.0001
Documented as mental health disorder	1033 (28)	1041 (25)	904 (22)	634 (24)	3612 (25)	14,602 (100)	<0.0001

**Table 3 ijerph-19-10761-t003:** Receipt of primary care services in the cohort with established CVD by age.

	<65 (*n* = 3666)	65–74 (*n* = 4229)	75–84 (*n* = 4108)	≥85 (*n* = 2599)	Total (*n* = 14602)	*p*-Value
Components of Chronic Disease Management Plans						
Preparation of GPMP	1296 (35)	1939 (46)	2101 (51)	1046 (40)	6382 (44)	<0.0001
Coordination of TCA	1093 (30)	1659 (39)	1808 (44)	938 (36)	5498 (38)	<0.0001
Review of GPMP or TCA	963 (26)	1503 (36)	1709 (42)	901 (35)	5076 (35)	<0.0001
Preparation of GPMP & Review of GPMP or TCA	797 (22)	1245 (29)	1425 (35)	726 (28)	4193 (29)	<0.0001
Mental health care items						
Preparation of a GP Mental Health Treatment Plan *	231 (22)	145 (14)	71 (8)	19 (3)	466 (13)	<0.0001
Review of a GP Mental Health Treatment Plan *	94 (9)	52 (5)	29 (3)	5 (1)	180 (5)	<0.0001
GP Mental Health Treatment Consultation *	153 (15)	116 (11)	98 (11)	31 (5)	398 (11)	<0.0001
Guideline recommended prescribed cardiovascular medication						
Blood pressure-lowering medication	2162 (59)	3001 (71)	3034 (74)	1783 (69)	9980 (68)	<0.0001
Lipid-lowering medication	2155 (59)	2685 (63)	2540 (62)	1137 (44)	8517 (58)	<0.0001
Antiplatelet medication	1772 (48)	2178 (52)	1985 (48)	1069 (41)	7004 (48)	<0.0001
Combination of blood pressure-lowering medication, lipid-lowering medication and antiplatelet medication	1113 (30)	1494 (35)	1333 (32)	545 (21)	4485 (31)	<0.0001
Flu vaccination						
	1982 (54)	3317 (78)	3532 (86)	2190 (84)	11,021 (76)	<0.0001

GP, General Practitioner; GPMP, General Practice Management Plan; TCAs, Team Care Arrangements; * Denominator: those who were documented as having mental health disorder.

**Table 4 ijerph-19-10761-t004:** Multiple-adjusted Odds Ratios by age group (reference group < 65 years) for the receipt of MBS items and guideline-recommended prescription medications and achievement of control targets in the cohort with established CVD.

Outcomes	Age (years)	Odds Ratio (95% Confidence Interval)	*p*-Value
Components of Chronic Disease Management Plans			
Preparation of GPMP	65–74	1.6 (1.33, 1.91)	0.0026
	75–84	1.88 (1.49, 2.36)	
	≥85	1.55 (1.11, 2.16)	
Coordination of TCA	65–74	1.49 (1.27, 1.75)	0.0029
	75–84	1.8 (1.43, 2.25)	
	≥85	1.65 (1.17, 2.33)	
Review of GPMP or TCA	65–74	1.63 (1.43, 1.86)	0.0014
	75–84	2.09 (1.64, 2.66)	
	≥85	1.93 (1.39, 2.7)	
Mental health care items			
Preparation of a GP Mental Health Treatment Plan *	65–74	0.55 (0.38, 0.82)	0.0005
	75–84	0.3 (0.21, 0.45)	
	≥85	0.17 (0.11, 0.27)	
Review of a GP Mental Health Treatment Plan *	65–74	0.51 (0.36, 0.74)	0.0010
	75–84	0.34 (0.19, 0.61)	
	≥85	0.11 (0.05, 0.23)	
GP Mental Health Treatment Consultation *	65–74	0.69 (0.49, 0.96)	0.0481
	75–84	0.69 (0.49, 0.99)	
	≥85	0.49 (0.31, 0.76)	
Guideline recommended prescribed cardiovascular medication			
Blood pressure-lowering medication	65–74	1.58 (1.33, 1.88)	0.0024
	75–84	1.92 (1.55, 2.37)	
	≥85	1.8 (1.34, 2.42)	
Lipid-lowering medication	65–74	1.06 (0.94, 1.18)	0.0141
	75–84	0.92 (0.83, 1.02)	
	≥85	0.54 (0.44, 0.67)	
Antiplatelet medication	65–74	1.02 (0.88, 1.17)	0.0421
	75–84	0.87 (0.74, 1.03)	
	≥85	0.74 (0.58, 0.93)	
Combination of blood pressure-lowering medication, lipid-lowering medication and antiplatelet medication	65–74	1.12 (0.93, 1.35)	0.0183
	75–84	0.93 (0.71, 1.22)	
	≥85	0.64 (0.44, 0.94)	
Flu vaccination			
	65–74	3.13 (2.6, 3.77)	<0.0001
	75–84	6.19 (5.19, 7.39)	
	≥85	8.23 (6.45, 10.49)	

Adjusted for age groups, gender, Indigenous status, current smoker, type 2 diabetes, systolic blood pressure, total cholesterol, body mass index categories and alcohol consumption. * Denominator: those who was documented as having mental health disorder.

## Data Availability

The data presented in this study are available on request from the corresponding author.

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
