# Peer review of "Age-Related Variation in the Provision of Primary Care Services and Medication Prescriptions for Patients with Cardiovascular Disease"

_ijerph, 2022, doi:10.3390/ijerph191710761_

Round 1
Reviewer 1 Report
Thanks for this study. Research on (correct) healthcare utilization among elderly people is highly relevant in a world that is increasingly ageing. In your study, you performed a secondary analysis using an impressive amount of data that was collected for a cluster randomized controlled trial. The results are very interesting, but extra caution is imminent when interpreting the data, since these were not collected in order to answer your current research question. Furthermore, although a great wealth of data were available for the current analysis- it is quite specific for the situation in Australia, and therefore hampers generalization.
Could you clarify the following:
* Line 120: item (ii) a documented dx of CVD- this requires more specification (e.g. which ICD codes).
* Statistical methods:
- Could you specify which variables were tested using Cochran-Armitage trend test (and not!! Cochrane), and why this test was chosen. This is typically a test for trend- and is therefore a bit unusual for a cross-sectional design with a single measurement per participant.
- More details should be included in lines 162-64: which method of entering (forward/backward stepwise) was used, how was the final model chosen, which measure for goodness of fit was chosen. Certainly- this should be added to the results section too.
* Interestingly, you do not mention the point of generalization beyond Australia in paragraph 5- I think you should. How can these findings contribute to non-Australian healthcare settings?
In order to improve the readability:
* Lines 72-89: this is really hard to read for someone not familiar with the Australian healthcare system. Could you add a table outlining the different benefit plans.
* There are really a lot of abbreviations in the manuscript; with (intriguingly) lists of very common/standard abbreviations (e.g. SBP, LDL, etc.). Could you add a full list of all abbreviations at the beginning of the manuscript?
Reviewer 2 Report
Title: “Age-related variation in the provision of primary care services and medication prescriptions for patients with cardiovascular disease”
General Comments: This study used a retrospective cross-sectional study design, conducting secondary data analysis from a cluster randomized controlled trial that recruited 50 primary care practices across four states in Australia. The study aim was to evaluate age-related variation in the utilization of primary care services including chronic disease management plans, mental health care, prescription of guideline indicated medications and influenza vaccination among patients with cardiovascular disease (CVD) presenting to primary care. A total of 14,602 patients with CVD were stratified into four age groups: adults <65 years, youngest-old (65–74 years), middle-old (75–84 years), and oldest-old (≥85 years) for the analyses. This is a well written manuscript.
Specific suggestions for revision:
Results
Lines 212 and 214, suggest adding the word “health” as follows: “…mental health disorder…”
Discussion
Lines 298 and 306, suggest adding the word “health” as follows: “…mental health disorder…”
Thank you for the invitation to review this manuscript!
Reviewer 3 Report
Dear Authors
Strengths
The manuscript have many strengths (novelty, interest to readers and quality of the presentation).
Limitations
1- The authors document already the limitations of the study at the end of the manuscript.
2- Could the authors documents how many patients continue from the start till the age more than 85 (compare the patient regarding the items discussed at young age and old if it is accessible).
3- Could it be accessible to compare the male and female regarding to age variations .
Round 2
Reviewer 1 Report
Dear authors,
Thanks for your clarifications.
I am looking forward to the publication!